# AR-DIFFUSION: Auto-Regressive Diffusion Model for Text Generation

**Tong Wu**[1][*][†]**, Zhihao Fan**[2][*][†]**, Xiao Liu**[3]**, Hai-Tao Zheng**[1,8][‡]**, Yeyun Gong**[3][‡]**, Yelong Shen**[4]**,**
**Jian Jiao**[5]**, Juntao Li**[6]**, Zhongyu Wei**[2]**, Jian Guo**[7][‡]**, Nan Duan**[3][‡]**, Weizhu Chen**[4][‡]

[1]Shezhen International Graduate School, Tsinghua University, [2] Fudan University,
[3]Microsoft Research Asia, [4]Microsoft Azure AI, [5]Microsoft,
[6]Soochow University, [7]IDEA Research, [8]Pengcheng Laboratory
{yegong, yeshe, nanduan, wzchen}@microsoft.com,
zheng.haitao@sz.tsinghua.edu.cn, guojian@idea.edu.cn

## Abstract

Diffusion models have gained significant attention in the realm of image generation due to their exceptional performance. Their success has been recently expanded to text generation via generating all tokens within a sequence concurrently. However, natural language exhibits a far more pronounced sequential dependency in comparison to images, and the majority of existing language models are trained with a left-to-right auto-regressive approach. To account for the inherent sequential characteristic of natural language, we introduce Auto-Regressive Diffusion (AR-DIFFUSION). AR-DIFFUSION ensures that the generation of tokens on the right depends on the generated ones on the left, a mechanism achieved through employing a dynamic number of denoising steps that vary based on token position. This results in tokens on the left undergoing fewer denoising steps than those on the right, thereby enabling them to generate earlier and subsequently influence the generation of tokens on the right. In a series of experiments on various text generation tasks, including text summarization, machine translation, and common sense generation, AR-DIFFUSION clearly demonstrated its superiority over existing diffusion language models and that it can be $100\times \sim 600\times$ faster when achieving comparable results. Our code is available at this https URL.

## 1 Introduction

Text generation is a fundamental task within the field of natural language processing (NLP). Pre-trained language models like GPT-4 [OpenAI, 2023], LLaMA [Touvron et al., 2023], and Alpaca [Taori et al., 2023] have garnered significant attention with their ability to generate fluent and human-like textual content. These models utilize the auto-regressive (AR) Transformer decoders [Vaswani et al., 2017] to emit generated tokens one-by-one in sequential order from left to right. By leveraging the power of position dependency, AR models are able to enhance the naturalness, coherence, and adherence to human language conventions in the generated text [Brown et al., 2020].

Recent studies have shown the remarkable performance of diffusion models in image generation [Ho et al., 2020], motivating researchers to extend diffusion to text generation [Li et al., 2022a, Gong et al., 2022, Dieleman et al., 2022, Yuan et al., 2022, Ye et al., 2023]. By introducing timestep, these methods progressively regulate the interpolation between the original tokens and Gaussian noise, then iteratively denoise for text generation. At each timestep, the diffusion-based text generator predicts

---

[*]Work done during an internship at Microsoft Research Asia.
[†]These authors contributed equally to this work.
[‡]Corresponding author.

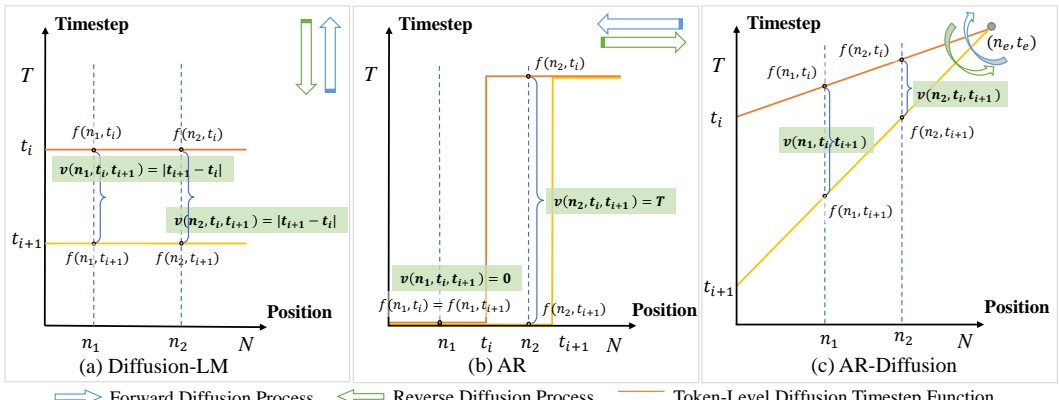

Figure 1: Model behaviors illustrated on a two-dimensional coordinate system, where the horizontal axis stands for the position and the vertical axis represents the diffusion timestep. In the inference stage, different models will behave differently. (a) For the typical Diffusion-LM [Li et al., 2022a], each token share the identical movement speed $v(n_1, t_i, t_{i+1}) = v(n_2, t_i, t_{i+1}) = |t_{i+1} - t_i|$. (b) For AR from the perspective of diffusion models, the tokens have two states based on the degree of interpolation between the original tokens and Gaussian noise: to be decoded (at timestep $t = T$) and already decoded (at timestep $t = 0$). Specifically, we have $v(n_1, t_i, t_{i+1}) = 0$ and $v(n_2, t_i, t_{i+1}) = T$. (c) In AR-DIFFUSION, $(n_e, t_e)$ is the coordinate of anchor point. Tokens in different positions exhibit varying movement speeds, such as $v(n_1, t_i, t_{i+1}) > v(n_2, t_i, t_{i+1})$ when $n_1 < n_2$.

all tokens simultaneously following Non-Auto-Regression (NAR) [Lewis et al., 2020, Qi et al., 2020, 2021, Li et al., 2022b], leading to faster decoding speed compared to AR. However, it also inherits the drawback of NAR, namely the sacrifice of inter-token position dependency [Li et al., 2022c] and the drop of generation performance [Bao et al., 2021].

To conduct a comprehensive analysis, we introduce a two-dimensional coordinate system to track the diffusion timestep of tokens $f(\cdot)$ positioned at various locations. As illustrated in Figure 1, the system assigns the token position $n \in [1, N]$ to the horizontal axis and the diffusion timestep $t \in [0, T]$ to the vertical axis. Diffusion-LM [Li et al., 2022a], which is followed by existing diffusion-based text generation models, is shown in Figure 1(a). It assigns a uniform timestep $t$ to all tokens. In contrast, tokens in the AR model depicted in Figure 1(b) exhibit distinct timesteps within a generation step $(t_i)$. For instance, the already decoded token at position $n_1$ has a timestep of 0, while the to-be-decoded token at position $n_2$ has a timestep of $T$. This approach effectively captures the sequential dependency. Motivated by this observation, we introduce AR-DIFFUSION, an auto-regressive diffusion method, for the disparity in token positions and the principle of sequential token identification.

In AR-DIFFUSION, we propose a **multi-level diffusion strategy** that includes both sentence-level and token-level diffusion. We randomly choose a sentence-level timestep $t$, and assign **dynamic movement speeds** $v(\cdot)$ by determining position-sensitive token-level timestep $f(n, t)$ for each token. This enables tokens at the left of a sentence to undergo faster movement from random Gaussian noise to token embedding, while those at the right of the sentence experience slower movement to better utilize information from previously denoised tokens. During inference, to reduce the significant number of inference steps (e.g., 2,000) required in Diffusion-LM [Li et al., 2022a], SeqDiffSeq [Yuan et al., 2022] and GENIE [Lin et al., 2023], we introduce a skipping mechanism that collaborates with the multi-level diffusion strategy to accelerate the process.

Experimental results across various text generation tasks, such as text summarization, machine translation, and common sense generation, have consistently demonstrated that AR-DIFFUSION surpasses existing text diffusion models, including AR methods in terms of both quality and diversity. Moreover, our verification reveals that AR-DIFFUSION requires fewer resources during decoding while maintaining superior performance. It achieves $100\times$ faster than SeqDiffSeq [Yuan et al., 2022] in machine translation and $600\times$ faster than GENIE [Lin et al., 2023] in text summarization while delivering comparable results. Furthermore, it demonstrates promising results even in a challenging scenario where decoding is limited to only two steps.

## 2 Preliminary

### 2.1 Conditional Generative Language Models

In the field of natural language generation, conditional generative models are commonly implemented using either auto-regressive (AR) or non-auto-regressive (NAR) methods. In AR [Vaswani et al., 2017], tokens on the right are predicted based on visible left tokens. The likelihood is given by $p_{\text{AR}}(\boldsymbol{y}|\boldsymbol{x}) = \prod_{i=1}^{N} p(\boldsymbol{y}_i|\boldsymbol{y}_{1:i-1}; \boldsymbol{x})$, where $y_i$ denotes the $i$-th token of $\boldsymbol{y}$. On the other hand, NAR [Gu et al., 2017] assumes conditional independence among tokens and generates them uniformly without distinction during decoding, resulting in the likelihood $p_{\text{NAR}}(\boldsymbol{y}|\boldsymbol{x}) = \prod_{i=1}^{N} p(\boldsymbol{y}_i|\boldsymbol{x})$. This parallel generation approach is of lower quality compared to AR, although it offers a substantial speed advantage.

### 2.2 Diffusion Models for Text Generation

Recently, Li et al. [2022a] propose a natural language generation model based on the diffusion process, which is typically divided into a forward noising process and a reverse denoising process.

Specifically, the forward process is a fixed linear Gaussian model, which gradually perturbs the random variable $\boldsymbol{z}_0$ until it becomes the standard Gaussian distribution. This can be formalized as:

$$q(\boldsymbol{z}_t \mid \boldsymbol{z}_0; \boldsymbol{x}) = \mathcal{N}(\boldsymbol{z}_t; \sqrt{\bar{\alpha}_t}\boldsymbol{z}_0, (1 - \bar{\alpha}_t)\mathbf{I}), \tag{1}$$

where, $\bar{\alpha}_t = \prod_{i=1}^{t} \alpha_i$, and $\alpha_i$ is a coefficient that monotonically decreases with timestep $t$, $\boldsymbol{z}_t$ is the latent state at timestep $t$.

The reverse process is to initiate from standard Gaussian noise and progressively utilize the denoising transition $p_{\boldsymbol{\theta}}(\boldsymbol{z}_{t-1}|\boldsymbol{z}_t; \boldsymbol{x})$ for generation.

$$p_{\boldsymbol{\theta}}(\boldsymbol{z}_{t-1} \mid \boldsymbol{z}_t; \boldsymbol{x}) = \mathcal{N}(\boldsymbol{z}_{t-1}; \mu_{\boldsymbol{\theta}}(\boldsymbol{z}_t, t; \boldsymbol{x}), \Sigma_{\boldsymbol{\theta}}(\boldsymbol{z}_t, t; \boldsymbol{x})), \tag{2}$$

where the mean $\mu_{\boldsymbol{\theta}}$ and variance $\Sigma_{\boldsymbol{\theta}}$ are learned from the model. In particular, we follow Li et al. [2022a]'s approach of using predefined variance without trainable parameters.

To extend the continuous diffusion process to discrete text generation, Li et al. [2022a] introduce an additional Markov transition from the discrete tokens $\boldsymbol{y}$ to the latent variable $\boldsymbol{z}_0$. In practice, we add an embedding step $q_\phi(\boldsymbol{z}_0|\boldsymbol{y}) = \mathcal{N}(\boldsymbol{z}_0; \text{Emb}(\boldsymbol{y}), (1 - \alpha_0)\mathbf{I})$ in the forward process, and use a trainable rounding step which is parametrized by $p_{\boldsymbol{\theta}}(\boldsymbol{y}|\boldsymbol{z}_0; \boldsymbol{x}) = \prod_{i=1}^{N} p_{\boldsymbol{\theta}}(y_i|z_0^i; \boldsymbol{x})$ in the reverse process. In each timestep, we utilize an encoder-decoder model $\boldsymbol{g}_{\boldsymbol{\theta}}(\boldsymbol{z}_t, t; \boldsymbol{x})$ to approximate $\boldsymbol{z}_0$ [Lin et al., 2023] in a NAR manner and then estimate $\mu_{\boldsymbol{\theta}}(\boldsymbol{z}_t, t; \boldsymbol{x})$.

In consequence, combined with maximizing the evidence lower bound (ELBO) of $\log p_{\boldsymbol{\theta}}(\boldsymbol{y}|\boldsymbol{x})$, our training objective of the conditional diffusion language model is:

$$\mathcal{L} = \mathbb{E}_{q_\phi(\boldsymbol{z}_{0:T}|\boldsymbol{y})} \left[ -\log p_{\boldsymbol{\theta}}(\boldsymbol{y} \mid \boldsymbol{z}_0; \boldsymbol{x}) + \sum_{t=1}^{T} \|\boldsymbol{z}_0 - \boldsymbol{g}_\theta(\boldsymbol{z}_t, t; \boldsymbol{x})\|^2 \right]. \tag{3}$$

## 3 Methodology

### 3.1 Multi-Level Diffusion

In the typical diffusion process, every token in the text sequence has the same diffusion timestep. In order to leverage the sequential nature of language, we enable tokens to have different diffusion timesteps during the forward and reverse pass. To accomplish this, we propose a multi-level diffusion strategy that includes both sentence-level and token-level diffusion. Firstly, at the sentence-level, we follow Diffusion-LM [Li et al., 2022a] to randomly select a timestep $t$. Secondly, at the token-level, we incorporate positional information $n \in [1, N]$ based on the sentence-level timestep to regulate the diffusion timestep for the current token. The procedure is illustrated as:

$$\boldsymbol{z}_t = (\boldsymbol{z}_{f(1,t)}^1, \boldsymbol{z}_{f(2,t)}^2, \cdots, \boldsymbol{z}_{f(N,t)}^N), \tag{4}$$

where $N$ is the given target sentence length, $\boldsymbol{z}_t$ is the sentence representation at timestep[4] $t$, $\boldsymbol{z}_{f(n,t)}^n$ is the latent representation for the $n$-th token at sentence-level timestep $t$, and $f(n, t)$ is a token-level

---

[4]Please note that if we talk about a "timestep" without explicitly indicating that it is for token-level, it should be for sentence-level.

timestep function that denotes the token-level diffusion timestep determined by token position $n$ and sentence-level timestep $t$.

We visualize the token-level timestep $(n, f(n, t))$ onto a two-dimensional coordinate system as Figure 1, which takes the token **position** as the horizontal axis and the sentence-level **timestep** as the vertical axis. Furthermore, to provide a more profound description of the characteristics of movement, we define the speed of movement as the following equation.

$$v(n, t_i, t_{i+1}) = f(n, t_{i+1}) - f(n, t_i), \tag{5}$$

where $t_i$ and $t_{i+1}$ are the start and end sentence-level diffusion timesteps. It can be observed that tokens in Diffusion-LM share the same movement speed, while those in AR exhibit different speeds.

### 3.2 Token-Level Diffusion with Dynamic Movement Speed

Based on the speed of movement, we propose a fundamental principle, dynamic movement speed, for designing the token-level diffusion timestep function $f(n, t)$ to take advantage of AR in diffusion. Specifically, elements on the left side of a sentence undergo higher movement speed from random Gaussian noise to token embedding, while those on the right side experience lower movement speed, thereby they can be generated in the later sentence-level timestep and utilize information from previously generated tokens more effectively.

---

**Algorithm 1** Training Process of AR-DIFFUSION.

---

**Input**: Dataset $\{(\boldsymbol{x}, \boldsymbol{y})\}$, maximum timestep number $T$ and maximum target length $N$.
**Output**: Optimized model parameters $\boldsymbol{\theta}$.
1: Define an anchor point $(n_e, t_e)$[5].
2: **repeat**
3:  Sample $(\boldsymbol{x}, \boldsymbol{y})$ from the dataset and embed $\boldsymbol{y}$ into $\boldsymbol{z}_0$.
4:  Sample a sentence-level timestep $t$ from the interval $[0, N + T]$, then the start point is determined by the following equation:

$$(n_s, t_s) = \big(\text{clip}(N - t, 0, N), \text{clip}(t - N, 0, T)\big) \tag{6}$$

5:  Use the point-slope linear function to determine the token-level timestep $f(n, t)$ in position $n$:

$$f(n, t) = \text{clip}\Big(\frac{t_e - t_s}{n_e - n_s}(n - n_s) + t_s, 0, T\Big) \tag{7}$$

6:  Sample $\boldsymbol{z}_{f(n,t)}^n$ for each $n$ in different positions with Gaussian reparameterization.
7:  According to equation (3) and equation (9), employ gradient descent to optimize the objective:

$$\min_{\theta} \Big[ - \log p_{\boldsymbol{\theta}}(\boldsymbol{y} \mid \boldsymbol{z}_0; \boldsymbol{x}) + \sum_{n=1}^{N} \big\| \boldsymbol{g}_{\boldsymbol{\theta}}(\boldsymbol{z}_{f(n,t)}^n, f(n,t); \boldsymbol{x}) - \boldsymbol{z}_0 \big\|^2 \Big] \tag{8}$$

8: **until** converged

---

Following the guidance of the principle, we develop a token-level diffusion strategy with the linear function, which is shown in Figure 1(c). In particular, the procedure is illustrated in Algorithm 1, where $\text{clip}(x, \min, \max)$ function is to clamp all elements in $x$ into the range $[\min, \max]$. Specifically, in the forward process of diffusion, the start point goes to the left from $(N, 0)$ to $(0, 0)$ along the horizontal axis and then moves up to $(0, T)$ along the vertical axis. Therefore, the entire range of sentence-level timestep is extended to $[0, N + T]$.

In the reverse diffusion process, the multi-level diffusion follows the formula:

$$\boldsymbol{g}_{\boldsymbol{\theta}}(\boldsymbol{z}_t, t; \boldsymbol{x}) = \boldsymbol{g}_{\boldsymbol{\theta}}\Big((\boldsymbol{z}_{f(1,t)}^1, f(1,t)), (\boldsymbol{z}_{f(2,t)}^2, f(2,t)), \cdots, (\boldsymbol{z}_{f(N,t)}^N, f(N,t)); \boldsymbol{x}\Big), \tag{9}$$

where $\boldsymbol{g}_{\boldsymbol{\theta}}(\boldsymbol{z}_{f(n,t)}^n, f(n,t); \boldsymbol{x})$ denotes the $n$-th element.

---

[5]In particular, the anchor point is set as $(2 \times N, T)$ in our implementation. The impact of different choices of the anchor point is discussed in supplementary material E.

### 3.3 Inference with Skipping

Typically, the generation process needs to go through all the sentence-level timesteps from $T + N$ to 0. To reduce the decoding time, we introduce a skipping mechanism that allows us to traverse a subset of timesteps.

---

**Algorithm 2** Inference Process of AR-DIFFUSION with the Skipping Mechanism.

---

    **Input**: Source condition $\boldsymbol{x}$, number of decoding steps $M$ and model parameters $\boldsymbol{\theta}$.
    **Output**: Predicted target embedding $\hat{\boldsymbol{y}}$.
1: Define an anchor point $(n_e, t_e)$.
2: Uniformly select a decreasing sequence of timesteps $\{t_i\}_{i=0}^{M}$ ranging from $T + N$ to 0.
3: Sample $\boldsymbol{z}_{t_0} \sim \mathcal{N}(\boldsymbol{0}, \mathbf{I})$.
4: **for** $i = 0$ to $M - 1$ **do**
5:     Calculate the start point $(n_s, t_s)$ using equation (6).
6:     Based on the current sentence-level inference steps $t_i$ and the next one $t_{i+1}$, assign token-level timesteps $f(n, t_i)$ and $f(n, t_{i+1})$ to token in position $n$ using equation (7).
7:     Reverse sample $\boldsymbol{z}_{t_{i+1}} = \left( \boldsymbol{z}_{f(1,t_{i+1})}^1, \boldsymbol{z}_{f(2,t_{i+1})}^2, \cdots, \boldsymbol{z}_{f(N,t_{i+1})}^N \right)$ from $p_\theta(\boldsymbol{z}_{t_{i+1}} \mid \boldsymbol{z}_{t_i}; \boldsymbol{x})$ with the following formulas:

$$p_\theta(\boldsymbol{z}_{t_{i+1}} \mid \boldsymbol{z}_{t_i}; \boldsymbol{x}) = \prod_{n=1}^{N} p_\theta\left( \boldsymbol{z}_{f(n,t_{i+1})}^n \mid \boldsymbol{z}_{f(n,t_i)}^n; \boldsymbol{x} \right) \tag{10}$$

$$p_\theta\left( \boldsymbol{z}_{f(n,t_{i+1})}^n \mid \boldsymbol{z}_{f(n,t_i)}^n; \boldsymbol{x} \right) \sim \mathcal{N}\left( \boldsymbol{z}_{f(n,t_{i+1})}^n; \lambda \boldsymbol{z}_{f(n,t_i)}^n + \mu \boldsymbol{g}_\theta(\boldsymbol{z}_{f(n,t)}^n, f(n,t); \boldsymbol{x}), \sigma \mathbf{I} \right) \tag{11}$$

8: **end for**
9: Map $\boldsymbol{z}_{t_M}$ to the nearest embedding $\hat{\boldsymbol{y}}$.

---

In practice, we propose an algorithm for the inference, illustrated in Algorithm 2.

$$\lambda = \frac{\sqrt{\frac{\bar{\alpha}_{f(n,t_i)}}{\bar{\alpha}_{f(n,t_{i+1})}}}(1 - \bar{\alpha}_{f(n,t_{i+1})})}{1 - \bar{\alpha}_{f(n,t_i)}}, \ \mu = \frac{\sqrt{\bar{\alpha}_{f(n,t_{i+1})}}(1 - \frac{\bar{\alpha}_{f(n,t_i)}}{\bar{\alpha}_{f(n,t_{i+1})}})}{1 - \bar{\alpha}_{f(n,t_i)}}, \ \sigma = \frac{(1 - \alpha_{f(n,t_i)})(1 - \bar{\alpha}_{f(n,t_{i+1})})}{1 - \bar{\alpha}_{f(n,t_i)}} \tag{12}$$

In equation (10), the conditional distribution of $\boldsymbol{z}_{t_{i+1}}$ is inferred by $p_\theta(\boldsymbol{z}_{t_{i+1}} | \boldsymbol{z}_{t_i}; \boldsymbol{x})$, and then we decompose it by positions due to the independent forward process of elements at different positions. From equation (11) to equation (12), we establish the relationship between tokens at different timesteps, and the detailed derivation can be found in supplementary material F.

## 4 Experiments

### 4.1 Tasks and Datasets

**Text Summarization**    This task involves taking a long document as input and generating a concise sentence as output. This requires models with the ability to identify important content and rewrite it in a condensed form. In our experiments, we use the publicly available XSUM [Narayan et al., 2018] and CNN/DAILYMAIL Hermann et al. [2015] on GLGE[6], which is also named as GLGE-Easy.

**Machine Translation**    Translation is a widely used sequence-to-sequence task. The input is a sequence of words in the source language, and the output is a sequence of corresponding words in the target language. We choose the IWSLT 2014 dataset and the data processing method is to follow the scripts provided by fairseq[7].

**Common Sense Generation**    In this task, the model is provided with a concept set consisting of objects and actions as input. The objective is to generate a sentence that incorporates these concepts and describes a realistic scenario. We use COMMONGEN[8] dataset for evaluation.

---

[6]https://microsoft.github.io/glge/
[7]https://github.com/facebookresearch/fairseq/tree/main/examples/translation
[8]https://inklab.usc.edu/CommonGen/

## 4.2 Main Results

The results on different datasets are shown in Table 1, Table 2, Table 3 and Table 4. The best result is **bolded** and the second-best result is underlined . "$k$" indicates the number of generated candidate samples[9]. It can be seen from the results in each table that AR-DIFFUSION achieves the best performance.

During the inference process, we utilize **20** inference steps and employ Minimum Bayes Risk (MBR) [Kumar and Byrne, 2004] decoding to select the best sample, following [Li et al., 2022a]. We choose MBR instead of the selection approach in GENIE, as GENIE picks up the best sample by calculating the maximum score for each generated one using ground truth, which introduces unfairness. To ensure a fair comparison, we re-implement GENIE using our configuration and perform inference with 20 steps. More experimental details can be found in the supplementary material B.

Table 1: Results on XSUM test set. The results of NAR and Semi-NAR are from Qi et al. [2021], and the results of AR are from GLGE [Liu et al., 2021].

| Methods | Pattern | ROUGE-1 | ROUGE-2 | ROUGE-L |
|---|---|---|---|---|
| NAT [Gu et al., 2017] | NAR | 24.0 | 3.9 | 20.3 |
| iNAT [Lee et al., 2018] | | 24.0 | 4.0 | 20.4 |
| CMLM [Ghazvininejad et al., 2019] | | 23.8 | 3.6 | 20.2 |
| LevT [Gu et al., 2019] | | 24.8 | 4.2 | 20.9 |
| InsT [Stern et al., 2019] | Semi-NAR | 17.7 | 5.2 | 16.1 |
| iNAT [Lee et al., 2018] | | 27.0 | 6.9 | 22.4 |
| CMLM [Ghazvininejad et al., 2019] | | 29.1 | 7.7 | 23.0 |
| LevT [Gu et al., 2019] | | 25.3 | 7.4 | 21.5 |
| LSTM [Greff et al., 2017] | AR[10] | 25.1 | 6.9 | 19.9 |
| Transformer [Vaswani et al., 2017] | | 30.5 | 10.4 | 24.2 |
| GENIE [Lin et al., 2023] ($k = 50$) | Diffusion | 29.3 | 8.3 | 21.9 |
| AR-DIFFUSION ($k = 50$) | | 31.7 | 10.1 | 24.7 |
| AR-DIFFUSION ($k = 500$) | | **32.2** | **10.6** | **25.2** |

Table 2: Results on CNN/DAILYMAIL test set. The results of AR are from GLGE Liu et al. [2021].

| Methods | Pattern | ROUGE-1 | ROUGE-2 | ROUGE-L |
|---|---|---|---|---|
| LSTM [Greff et al., 2017] | AR | 37.3 | 15.7 | 34.4 |
| Transformer [Vaswani et al., 2017] | | 39.5 | 16.7 | 36.7 |
| GENIE [Lin et al., 2023] ($k = 50$) | Diffusion | 34.4 | 12.8 | 32.1 |
| AR-DIFFUSION ($k = 50$) | | 39.6 | 16.3 | 37.1 |
| AR-DIFFUSION ($k = 500$) | | **40.2** | **17.1** | **37.7** |

**Text Summarization**    The results presented in Table 1 and Table 2 clearly demonstrate that AR-DIFFUSION outperforms the existing NAR and Semi-NAR approaches across all metrics. Moreover, AR-DIFFUSION consistently achieves significant improvements over GENIE in terms of all indicators. Furthermore, in comparison to Transformer, AR-DIFFUSION outperforms it on both ROUGE-1 and ROUGE-L, while achieving comparable performance in terms of ROUGE-2. Notably, when the sample number is 500, AR-DIFFUSION demonstrates superiority over Transformer across all the measures.

**Machine Translation**    Table 3 presents the BLEU score implemented by SeqDiffuSeq setting[11]. AR-DIFFUSION outperforms the non-auto-regressive CNAT in greedy search for a single sample, and achieves a substantial gain. Moreover, the BLEU score of AR-DIFFUSION surpasses GENIE by a large margin and shows a slightly better performance than the AR Transformer. Specially, AR-DIFFUSION achieves a more powerful result at $k = 500$.

---

[9]The relationship between sample number and results is discussed in supplementary material E.

[10]Notably, although AR's beam search has a small beam, the search space may be larger than 50 or even 500.

[11]We also report SacreBLEU in supplementary material C to compare with DINOISER.

Table 3: Results on IWSLT14 DE→EN test set following the setting of SEQDIFFUSEQ. "NFE" indicates the **N**umber of **F**unction **E**valuations [Ye et al., 2023].

| Methods | Pattern | BLEU | Steps | NFE (Steps×$k$) |
|---|---|---|---|---|
| Transformer [Vaswani et al., 2017] | AR | 34.74 | - | - |
| CNAT [Bao et al., 2021] | NAR | 29.81 | - | - |
| SeqDiffSeq [Yuan et al., 2022] ($k = 1$) | Diffusion | 29.83 | 2,000 | 2,000 (2,000 × 1) |
| AR-DIFFUSION ($k = 1$) | | 30.19 | 20 | 20 (20 × 1) |
| GENIE [Lin et al., 2023] ($k = 50$) | | 30.08 | 20 | 1,000 (20 × 50) |
| AR-DIFFUSION ($k = 50$) | Diffusion | 34.95 | 20 | 1,000 (20 × 50) |
| AR-DIFFUSION ($k = 500$) | | **35.62** | 20 | 10,000 (20 × 500) |

Table 4: Results on COMMONGEN dev set. Results of NAR and AR are from Lin et al. [2020].

| Methods | Pattern | ROUGE-2/L | | BLEU-3/4 | | METEOR | SPICE |
|---|---|---|---|---|---|---|---|
| bRNN-CopyNet [Gu et al., 2016] | | 9.23 | 30.57 | 13.60 | 7.80 | 17.40 | 16.90 |
| Trans-CopyNet [Lin et al., 2020] | AR | 11.08 | 32.57 | 17.20 | 10.60 | 18.80 | 18.00 |
| MeanPooling-CopyNet [Lin et al., 2020] | | 11.36 | 34.63 | 14.80 | 8.90 | 19.20 | 20.20 |
| LevT [Gu et al., 2019] | NAR | 12.22 | 35.42 | 23.10 | 15.00 | 22.10 | 21.40 |
| ConstLeven [Susanto et al., 2020] | | 13.47 | 35.19 | 21.30 | 12.30 | **25.00** | 23.20 |
| GENIE [Lin et al., 2023] ($k = 50$) | Diffusion | 12.89 | 35.21 | 22.00 | 13.30 | 24.30 | 23.00 |
| AR-DIFFUSION ($k = 50$) | | **13.93** | **37.36** | **25.60** | **16.40** | 25.00 | **24.20** |

**Common Sense Generation**   As depicted in Table 4, AR-DIFFUSION achieves superior performance compared to the current AR, NAR, and other diffusion methods across all the metrics on the COMMONGEN dataset.

Table 5: Experimental results of GENIE and AR-DIFFUSION with inference steps of **2** and **3** on XSUM test set. Take $k = 10$ to apply the MBR decoding strategy. (·) indicates the **drop** score compared to the 20-step.

| Methods | Steps | NFE | ROUGE-1 | ROUGE-2 | ROUGE-L | AVG Drop |
|---|---|---|---|---|---|---|
| GENIE | 2,000 | 20,000 | 30.36 | 8.78 | 23.31 | - |
| | 20 | 200 | 28.33 | 7.46 | 21.15 | - |
| | 3 | 30 | 25.03 (-3.30) | 5.32 (-2.14) | 18.17 (-2.98) | 2.81 |
| | 2 | 20 | 23.45 (-4.88) | 3.95 (-3.51) | 16.94 (-4.21) | 4.20 |
| AR-DIFFUSION | 20 | 200 | 30.99 | 9.32 | 23.95 | - |
| | 3 | 30 | 30.23 (-0.76) | 8.68 (-0.64) | 23.43 (-0.52) | **0.64** |
| | 2 | 20 | 29.28 (-1.71) | 7.99 (-1.33) | 22.98 (-0.97) | **1.34** |

### 4.3   Inference Efficiency

First, we use the number of function evaluations (NFE) as a measure to compare inference efficiency [Ye et al., 2023] in machine translation. From Table 3, it is evident that even when the NFE is reduced to 1% of SeqDiffSeq (equivalent to $100\times$ faster), AR-DIFFUSION still outperforms SeqDiffSeq. Moreover, increasing the number of generated candidate samples ($k = 500$) leads to further performance improvements, albeit with increased time consumption.

Second, we conduct experiments with an **extremely limited number of inference steps** (2 and 3)[12] and compare the performance with that of GENIE in XSUM. The results are presented in Table 5. When reducing the number of steps to 2, GENIE experiences a significant decline, with an average score of 4.20 in the AVG Drop column, while AR-DIFFUSION exhibits a comparatively smaller

---

[12]The time consumed by each step in the inference process is exactly the same.

decrease of 1.34. Furthermore, with 3 steps, although the performance deterioration of GENIE is somewhat reduced, the average score still shows a decline of 2.81. In contrast, AR-DIFFUSION maintains a high performance level, with an average score differing from the 20-step result by only 0.64. Notably, the results of AR-DIFFUSION at 3 steps are comparable to the results of GENIE at 2,000 steps. Therefore, compared to GENIE, the inference speed of AR-DIFFUSION can be accelerated by up to $600\times$.

Table 6: Diversity of **10** generated samples on XSUM test set and average of **10** results. The results of BART and GENIE are quoted from Lin et al. [2023].

| Methods | BART | | | | | | GENIE | AR-DIFFUSION |
|---|---|---|---|---|---|---|---|---|
| **Sampling** | Greedy Search | Beam Search | Diverse Beam Search | Typical Sample | Top-k Sample | Nucleus Sample | Diffusion | |
| SELF-BLEU ↓ | 100.0 | 93.4 | 75.6 | 76.9 | 80.2 | 79.1 | 29.3 | 30.4 |

## 4.4 Analysis

**Diversity of Samples** Diversity is a key advantage of diffusion models. To measure the diversity of generated samples, We adopt the SELF-BLEU [Zhu et al., 2018] metric, in which a lower score indicates higher diversity. In Lin et al. [2023], various sampling methods were applied to the pre-trained auto-regressive model BART[13]. As shown in Table 6, AR-DIFFUSION achieves significantly higher diversity compared to the auto-regressive model. Furthermore, the diversity can be comparable to GENIE with a better performance.

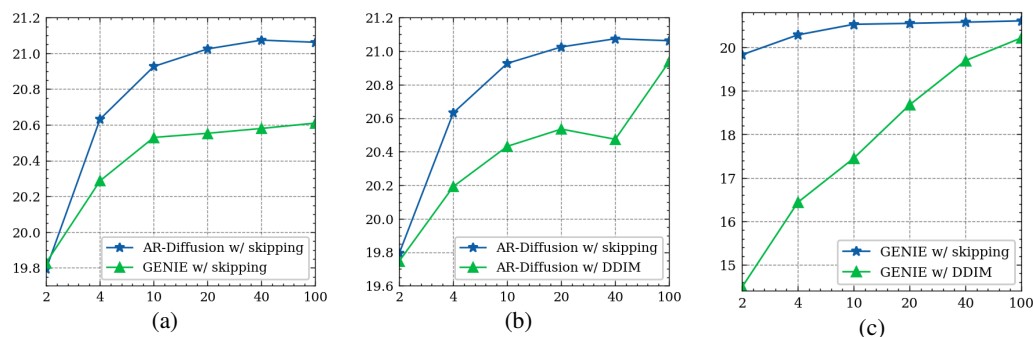

Figure 2: Ablation experiments on XSUM test set and taking $k = 5$. The horizontal axis is the number of inference steps and the vertical axis is AVG-ROUGE = (ROUGE-1 + ROUGE-2 + ROUGE-L) / 3.

**Ablation Study** To demonstrate the effectiveness of our proposed method, we perform ablation experiments on the XSUM dataset. Our results show that both our proposed multi-level diffusion and skipping mechanism are essential for achieving the high performance of AR-DIFFUSION.

Maintaining the skipping inference method, we remove the token-level diffusion during the training process, which degenerates to GENIE w/ skipping. The comparison results are shown in Figure 2(a). It can be observed that after removing, the AVG-ROUGE score is greatly lower after 2 steps.

The performance of applying our proposed skipping mechanism and DDIM [Song et al., 2021] to AR-DIFFUSION is shown in Figure 2(b). The results demonstrate that the skipping mechanism consistently outperforms DDIM in various inference steps. Additionally, the skipping mechanism can be easily applied to GENIE. As depicted in Figure 2(c), DDIM suffers a significant drop in performance when the number of inference steps is less than 40. In contrast, the skipping mechanism consistently maintains good performance across all inference steps.

---

[13]Specific details are written in supplementary material G.

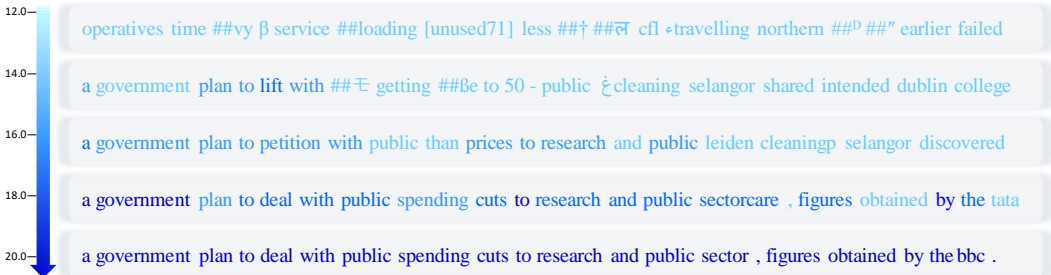

Figure 3: The intermediate state of AR-DIFFUSION gradually generating real text from a standard Gaussian noise through 20 steps. The brightness of the color represents the magnitude of the logits, with darker colors indicating larger logits. More cases are shown in the supplementary materials H.

**Case Study** By mapping the state to the token with the highest logits, we visualize the intermediate states of AR-DIFFUSION. As depicted in Figure 3, AR-DIFFUSION undergoes a denoising process, transforming the random Gaussian noise into a coherent sentence over 20 steps, and we present 5 of them. With the progression of each timestep, compared to the tokens on the right side of the sentence, the tokens on the left side demonstrate faster determination and a rapid increase in the corresponding logits. This behavior is consistent with our principle of dynamic movement speed from left to right.

## 5 Related Work

**AR and NAR Language Models** AR models have been the dominant approach for text generation [OpenAI, 2023, Touvron et al., 2023, Dong et al., 2023], but their token-by-token generation nature often leads to unsatisfactory inference speed. To address this issue, NAR models have been developed in recent years. The NAR method is initially proposed by Gu et al. [2017], its objective is generate the entire output sequence in parallel, thereby improving generation speed and efficiency. Subsequently, LevT [Gu et al., 2019] adopts insertion and deletion to address the lack of flexibility in NAR generation, CMLM [Ghazvininejad et al., 2019] utilizes a masked language model to improve the quality of NAR generation through a constant number of iterations, and CNAT [Bao et al., 2021] introduces latent variables to represent the category information of the target word to make full use of the latent representation. However, these NAR methods are hard to model inter-token position dependency and deficient in generation performance.

**Continuous Text Diffusion** The application of diffusion models to continuous text space is first introduced by Li et al. [2022a]. Through the embedding and rounding processes, the direct integration of continuous noise into word embeddings was accomplished. After that, more people attempt to adopt continuous text diffusion model to solve sequence-to-sequence tasks. DiffuSeq [Gong et al., 2022] divides the input into two parts, utilizing one part as a condition, and perturbs the other part with noise. CDCD [Dieleman et al., 2022] proposes score interpolation and time warping to allow diffusion model and Euclidean embedding to share the same loss function for training. SeqDiffSeq [Yuan et al., 2022], GENIE [Lin et al., 2023] and DINOISER [Ye et al., 2023] incorporate diffusion model into the encoder-decoder structure through cross-attention mechanisms.

It is important to highlight the differences between our method and both ARDMs [Hoogeboom et al., 2022] and TimeGrad [Rasul et al., 2021], despite the common references to autoregression and diffusion in all these. ARDMs employ an order-agnostic technique, leveraging masking and prediction for generation in arbitrary orders. On the other hand, TimeGrad integrates RNN and diffusion to model the conditional distribution of future steps of multivariate time series. In contrast, our research focuses on implementing the diffusion process within a continuous embedding space, with the primary aim of generating text in a left-to-right sequence.

## 6 Conclusion

This paper introduces AR-DIFFUSION, which exhibits AR-like generation behavior but enables efficient parallel decoding. Embracing the inherent sequential nature of language, we propose a multi-

level diffusion model, consisting of sentence-level and token-level components, to assign dynamic movement speeds to tokens. Consequently, compared to those on the right, the left tokens undergo fewer denoising steps and generate earlier to subsequently influence the later ones. Furthermore, we introduce a skipping mechanism to facilitate parallel generation within the multi-level diffusion framework. The experimental results across various tasks demonstrate that AR-DIFFUSION surpasses existing diffusion models in terms of quality while maintaining diversity. Additionally, compared to existing diffusion language models, AR-DIFFUSION achieves comparable results while being $100\times \sim 600\times$ faster.

## 7   Acknowledgements

This research is supported by National Natural Science Foundation of China (Grant No.62276154), Research Center for Computer Network (Shenzhen) Ministry of Education, the Natural Science Foundation of Guangdong Province (Grant No. 2023A1515012914), Basic Research Fund of Shenzhen City (Grant No. JCYJ20210324120012033 and JSGG20210802154402007), the Major Key Project of PCL for Experiments and Applications (PCL2021A06), and Overseas Cooperation Research Fund of Tsinghua Shenzhen International Graduate School (HW2021008).

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

# A   Limitation

A primary limitation of our work lies in the requirement of generating a large number of candidate samples for optimal performance. As an illustration in Table 2 of CNN/DAILYMAIL dataset, AR-DIFFUSION ($k = 50$) achieves a 0.8 lower ROUGE-2 score compared to AR-DIFFUSION ($k = 500$). We anticipate exploring more efficient sampling strategies to minimize the number of generated samples without performance drop.

# B   Experimental Details

**Model Setup**   Our model configuration is implemented based on Transformer-base [Vaswani et al., 2017]. In particular, For XSUM and CNN/DAILYMAIL, we set the diffusion embedding dimension to 128. For IWSLT14, we use 64-dimensional diffusion embedding, 4 attention heads and 1024-dimensional feed-forward layers. For COMMONGEN, we adopt 64-dimensional diffusion embedding, 8 attention heads and 512-dimensional feed-forward layers.

**Training and Inference**   In the training phase, we employ a square-root noise schedule and 2,000 diffusion steps [Li et al., 2022a]. Specially, we use the tokenizer and vocabulary constructed by Byte Pair Encoding (BPE)[14] [Kudo and Richardson, 2018] for translation tasks. For other tasks, we adopt the tokenizer and vocabulary of bert-base-uncased.

**Baselines**   We set four groups of baselines:

• NAR: NAT [Gu et al., 2017], iNAT [Lee et al., 2018], CMLM [Ghazvininejad et al., 2019], LevT [Gu et al., 2019] and CNAT [Bao et al., 2021];

• Semi-NAR: InsT [Stern et al., 2019], iNAT [Lee et al., 2018], CMLM [Ghazvininejad et al., 2019] and LevT [Gu et al., 2019];

• AR: bRNN [Gu et al., 2016], LSTM [Greff et al., 2017] and Transformer [Vaswani et al., 2017];

• Diffusion: DiffusionLM [Li et al., 2022a], CDCD [Dieleman et al., 2022], SeqDiffuSeq [Yuan et al., 2022], DINOISER [Ye et al., 2023] and GENIE [Lin et al., 2023].

**Metrics**   We follow the approach of Qi et al. [2020][15] to evaluate the **ROUGE-1/2/L** of the summarization task. For the evaluation of translation tasks, we adopt the setting of SeqDiffuSeq [Yuan et al., 2022] to report BLEU score. In addition, we also calculate the SacreBLEU score according to the setting of DINOISER [Ye et al., 2023] for comparison. For COMMONGEN, we employ ROUGE-2/L, BLEU-3/4, METEOR and SPICE under the evaluation methods of Lin et al. [2020][16].

Table 7: Training Parameter Settings. Batch Size = mini batch size $\times N_{gc} \times$ GPU number, Optimized Steps = total steps / $N_{gc}$, and $N_{gc}$ is gradient accumulation number.

| Dataset | Lr & Schedule | Batch Size | Optimized Steps | Target Length |
|---|---|---|---|---|
| XSUM | 8e-4 & Cosine | 128×3×8 | 80,000 / 3 | 50 |
| CNN/DAILYMAIL | 8e-4 & Cosine | 80×5×8 | 100,000 / 5 | 180 |
| IWSLT14 DE→EN | 2e-3 & Cosine | 192×2×8 | 160,000 / 2 | 90 |
| IWSLT14 EN→DE | 1.8e-3 & Cosine | 768×1×8 | 60,000 | 90 |
| COMMONGEN | 3e-4 & Constant | 384×1×8 | 40,000 | 54 |

**Training Parameters**   Our training parameters on different datasets are shown in Table 7. Our linear schedule warm up steps is $4,000 \times N_{gc}$, where $N_{gc}$ denotes gradient accumulation number. In addition, we use the AdamW (weight decay = 0.0) optimizer and dropout is 0.2. All experiments are implemented on 8 Tesla V100-32G. It takes about 20 hours to train XSUM and CNN/DAILYMAIL, about 5 hours to train IWSLT14, and about 2 hours to train COMMENGEN.

---

[14]We train bpe on the training set, and follow the vocabulary size of fairseq, IWSLT14 is set to 10,000 .

[15]https://github.com/microsoft/ProphetNet/tree/master/GLGE_baselines

[16]https://github.com/INK-USC/CommonGen/tree/master/evaluation/Traditional/eval_metrics

## C   More Results on IWSLT14

In Table 8 we give the `SacreBLEU` score according to the setting of DINOISER. AR-DIFFUSION has notable improvements over non-auto-regressive CMLM. Besides, AR-DIFFUSION achieves excellent performance among text diffusion models for both EN→DE and DE→EN tasks. Specifically, AR-DIFFUSION is far superior to GENIE and comparable to the newly proposed DINOISER at $n = 50$. Nevertheless, the performance is stronger than DINOISER when $k = 500$[17].

Table 8: `SacreBLEU` on the IWSLT14 test set. This result follows the setting of DINOISER.

| Methods | IWSLT14 | |
|---|---|---|
| | DE→EN | EN→DE |
| Transformer (AR, beam = 5) [Vaswani et al., 2017] | 33.61 | 28.30 |
| CMLM (NAR, $k$ = 5) [Ghazvininejad et al., 2019] | 29.41 | 24.34 |
| DiffusionLM ($k$ = 50) [Li et al., 2022a] | 29.11 | 22.91 |
| DINOISER ($k$ = 50) [Ye et al., 2023] | 31.61 | 26.14 |
| GENIE ($k$ = 50) [Lin et al., 2023] | 29.45 | 23.89 |
| AR-DIFFUSION ($k$ = 50) | 31.80 | 26.01 |
| AR-DIFFUSION ($k$ = 500) | **32.35** | **26.51** |

## D   Comparison of Runtime Speed

We also compare the runtime speeds of GENIE and AR-DIFFUSION by measuring the time taken for each sample generation, denoted as seconds per iteration (s/it). To ensure precision in our measurements, we sample 50 cases from the CNN/DAILYMAIL dataset. The generation process speed comprises two main components: function evaluation (FE, which is equivalent to the model forward computation) and MBR. All experiments are conducted on an A100-40G GPU and 50 CPUs.

The results are presented in Table 9. It is evident that for $k \leq 50$, the MBR duration is minimal and can be negligible relative to NFE. Furthermore, even when $k = 500$ and step = 20, AR-DIFFUSION remains approximately twice as efficient as GENIE ($k = 10$ and step = 2000).

Table 9: Comparison of inference speed between GENIE and AR-DIFFUSION.

| | GENIE | AR-DIFFUSION | | | |
|---|---|---|---|---|---|
| $k$ | 10 | 10 | 10 | 50 | 500 |
| Steps | 2000 | 3 | 20 | 20 | 20 |
| Speed of NFE (s/it) | 47.54 | 0.25 | 0.61 | 2.12 | 21.03 |
| Speed of MBR (s/it) | 0.02 | 0.02 | 0.02 | 0.08 | 7.14 |
| Total Speed (s/it) | **47.56** | **0.27** | **0.63** | **2.20** | **28.17** |
| `ROUGE-2` in XSUM | 8.78 | 8.68 | 9.32 | 10.1 | 10.6 |

## E   Impact of Minimum Bayes Risk and Anchor Point

**Minimum Bayes Risk**   To investigate the relationship between the number of generated candidate samples ($k$) and the quality of generation, we generate varying numbers of samples, ranging up to 1,000, on the IWSLT14 De→En test set and present the results in Figure 4. The curve demonstrates an initial gain of approximately 0.5 `SacreBLEU` within the first 200 samples, after which the gain becomes insignificant with generating more samples.

**Anchor Point**   We conduct experiments on AR-DIFFUSION using different anchor points $(n_e, t_e)$. These anchor points vary in terms of $n_e$ values, namely $1.0 \times N$, $2.0 \times N$ and $3.0 \times N$, where $N$ denotes the target sentence length. Additionally, they share a common $t_e$ value of $T$, which represents

---

[17]DINOISER has shown in their Figure 4 that their method is not better with a larger $k$.

the total time step of diffusion. We present the results in Table 10, and determine that the best result is achieved at $(n_e, t_e) = (2.0 \times N, T)$.

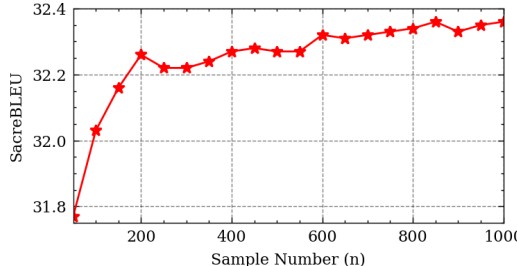

Figure 4: Relationship between the number of candidate samples for applying MBR and `SacreBLEU` on IWSLT14 DE→EN test set.

Table 10: Effect of anchor point at different positions on the IWSLT14 DE→EN test set. "$N$" indicates the target sequence length and "T" represents the total time step of diffusion.

| $n_e$ | $t_e$ | SacreBLEU |
|---|---|---|
| $1.0 \times N$ | $T$ | 31.23 |
| $2.0 \times N$ | $T$ | **31.80** |
| $3.0 \times N$ | $T$ | 31.58 |

## F    Proof of Inference with Skipping

During the inference process, skipping strategy requires the model $\boldsymbol{g}_\theta$ to infer the state $\boldsymbol{z}_{t_{i+1}}^{n_2}$ at a far-off timestep $t_{i+1}$ compared to the current state $\boldsymbol{z}_{t_i}^{n_2}$, where $t_{i+1} \ll t_i$. In our model, due to the dynamic speed setting, token $\boldsymbol{z}_{t_{i+1}}^{n_1}$ with smaller timestep $t_{i+1} \le t_i$, which is closer to $t_{i+1}$, and positions $n_1 \le n_2$ can provide stronger auxiliary information than $\boldsymbol{z}_{t_i}^{n_1}$. This reduces the difficulty of inferring states for tokens in the end, making our multi-level diffusion model particularly suitable for accelerating the generation process.

Through maximizing the evidence lower bound (ELBO) of $p(\boldsymbol{z}_0)$, the training object is equivalent to minimize the divergence between $q(\boldsymbol{z}_t|\boldsymbol{z}_{t-1}, \boldsymbol{z}_0)$ and $p_\theta(\boldsymbol{z}_{t-1}|\boldsymbol{z}_t)$ following [Luo, 2022].

By converting the joint probability distribution into a conditional probability distribution, we obtain the following formula for $q(\boldsymbol{z}_{t_{i+1}}|\boldsymbol{z}_{t_i}, \boldsymbol{z}_0)$.

$$
\begin{aligned}
q(\boldsymbol{z}_{t_{i+1}}|\boldsymbol{z}_{t_i}, \boldsymbol{z}_0) &= q(\boldsymbol{z}_{t_{i+1}}|\boldsymbol{z}_{t_i-1}, \boldsymbol{z}_{t_i}, \boldsymbol{z}_0)\, q(\boldsymbol{z}_{t_{i+1}-1}|\boldsymbol{z}_{t_i}, \boldsymbol{z}_0) \\
&= q(\boldsymbol{z}_{t_{i+1}}|\boldsymbol{z}_{t_i-1}, \boldsymbol{z}_0)\, q(\boldsymbol{z}_{t_{i+1}-1}|\boldsymbol{z}_{t_i}, \boldsymbol{z}_0) \\
&= q(\boldsymbol{z}_{t_{i+1}}|\boldsymbol{z}_{t_i-2}, \boldsymbol{z}_0)\, q(\boldsymbol{z}_{t_{i+1}-2}|\boldsymbol{z}_{t_i-1}, \boldsymbol{z}_0)\, q(\boldsymbol{z}_{t_{i+1}-1}|\boldsymbol{z}_{t_i}, \boldsymbol{z}_0) \\
&= \prod_{k=1}^{t_i - t_{i+1}} q(\boldsymbol{z}_{t_i-k}|\boldsymbol{z}_{t_i-k+1}, \boldsymbol{z}_0)
\end{aligned}
\tag{13}
$$

Similarly, we reach the same conclusion regarding $p_\theta(\boldsymbol{z}_{t_{i+1}}|\boldsymbol{z}_{t_i})$.

Based on equation (13), which consists of $q(\boldsymbol{z}_t|\boldsymbol{z}_{t-1}, \boldsymbol{z}_0)$, and the interchangeability between $q(\boldsymbol{z}_t|\boldsymbol{z}_{t-1}, \boldsymbol{z}_0)$ and $p_\theta(\boldsymbol{z}_{t-1}|\boldsymbol{z}_t)$, we can decompose $q(\boldsymbol{z}_{t_{i+1}}|\boldsymbol{z}_{t_i}, \boldsymbol{z}_0)$ by incorporating $\boldsymbol{z}_{t_i}$ and $\boldsymbol{z}_0$, and utilize our estimated $\boldsymbol{z}_0$ to determine the expression of $p_\theta(\boldsymbol{z}_{t_{i+1}}|\boldsymbol{z}_{t_i})$.

$$
q(\boldsymbol{z}_{t_{i+1}} \mid \boldsymbol{z}_{t_i}, \boldsymbol{z}_0) = \prod_{n=1}^{N} q\left(z_{f(n,t_{i+1})}^{n} \mid z_{f(n,t_i)}^{n}, z_0^{n}\right)
\tag{14}
$$

Next, we obtain the explicit expression $q\left(z_{f(n,t_{i+1})}^n \mid z_{f(n,t_i)}^n, z_0^n\right)$ through linear interpolation between $z_{f(n,t_i)}^n$ and $z_0^n$.

$$q\left(z_{f(n,t_{i+1})}^n \mid z_{f(n,t_i)}^n, z_0^n\right) = \frac{q(z_{f(n,t_i)}^n \mid z_{f(n,t_{i+1})}^n, z_0^n)q(z_{f(n,t_{i+1})}^n \mid z_0^n)}{q(z_{f(n,t_i)}^n \mid z_0^n)}$$

$$= \frac{\mathcal{N}\left(z_{f(n,t_i)}^n; \sqrt{\frac{\bar{\alpha}_{f(n,t_i)}}{\bar{\alpha}_{f(n,t_{i+1})}}} z_{f(n,t_{i+1})}^n, \left(1 - \frac{\bar{\alpha}_{f(n,t_i)}}{\bar{\alpha}_{f(n,t_{i+1})}}\right)I\right)\mathcal{N}\left(z_{f(n,t_{i+1})}^n; \sqrt{\bar{\alpha}_{f(n,t_{i+1})}}z_0^n, (1 - \bar{\alpha}_{f(n,t_{i+1})})I\right)}{\mathcal{N}\left(z_{f(n,t_i)}^n; \sqrt{\bar{\alpha}_{f(n,t_i)}}z_0^n, (1 - \bar{\alpha}_{t_i})I\right)}$$

$$\propto \exp\left\{-\frac{\left(z_{f(n,t_i)}^n - \sqrt{\frac{\bar{\alpha}_{f(n,t_i)}}{\bar{\alpha}_{f(n,t_{i+1})}}} z_{f(n,t_{i+1})}^n\right)^2}{2(1 - \frac{\bar{\alpha}_{f(n,t_i)}}{\bar{\alpha}_{f(n,t_{i+1})}})} - \frac{\left(z_{f(n,t_{i+1})}^n - \sqrt{\bar{\alpha}_{t_{i+1}}}z_0^n\right)^2}{1 - \bar{\alpha}_{f(n,t_{i+1})}}\right.$$

$$\left. + \frac{\left(z_{f(n,t_i)}^n - \sqrt{\bar{\alpha}_{f(n,t_i)}}z_0^n\right)^2}{1 - \bar{\alpha}_{f(n,t_i)}}\right\}$$

$$= \exp\left\{-\frac{1 - \bar{\alpha}_{t_i}}{2(1 - \frac{\bar{\alpha}_{t_i}}{\bar{\alpha}_{t_{i+1}}})(1 - \bar{\alpha}_{t_{i+1}})}\left[z_{f(n,t_{i+1})}^n{}^2 - 2\left(\frac{\sqrt{\frac{\bar{\alpha}_{f(n,t_i)}}{\bar{\alpha}_{f(n,t_{i+1})}}}(1 - \bar{\alpha}_{f(n,t_{i+1})})}{1 - \bar{\alpha}_{f(n,t_i)}}z_{f(n,t_i)}^n\right.\right.\right.$$

$$\left.\left.\left. + \frac{\sqrt{\bar{\alpha}_{f(n,t_{i+1})}}(1 - \frac{\bar{\alpha}_{f(n,t_i)}}{\bar{\alpha}_{f(n,t_{i+1})}})}{1 - \bar{\alpha}_{f(n,t_i)}}z_0^n\right)z_{f(n,t_{i+1})}^n\right]\right\}$$

$$\propto \mathcal{N}\left(z_{f(n,t_{i+1})}^n; \frac{\sqrt{\frac{\bar{\alpha}_{f(n,t_i)}}{\bar{\alpha}_{f(n,t_{i+1})}}}(1 - \bar{\alpha}_{f(n,t_{i+1})})}{1 - \bar{\alpha}_{f(n,t_i)}}z_{f(n,t_i)}^n + \frac{\sqrt{\bar{\alpha}_{f(n,t_{i+1})}}(1 - \frac{\bar{\alpha}_{f(n,t_i)}}{\bar{\alpha}_{f(n,t_{i+1})}})}{1 - \bar{\alpha}_{f(n,t_i)}}z_0^n,\right.$$

$$\left. \frac{(1 - \frac{\bar{\alpha}_{t_i}}{\bar{\alpha}_{t_{i+1}}})(1 - \bar{\alpha}_{t_{i+1}})}{1 - \bar{\alpha}_{t_i}}I\right)$$

$$= \mathcal{N}(z_{f(n,t_{i+1})}^n; \lambda z_{f(n,t_i)}^n + \mu z_0^n, \sigma I)$$

$$\tag{15}$$

where we have the following notations for simplification.

$$\lambda = \frac{\sqrt{\frac{\bar{\alpha}_{f(n,t_i)}}{\bar{\alpha}_{f(n,t_{i+1})}}}(1 - \bar{\alpha}_{f(n,t_{i+1})})}{1 - \bar{\alpha}_{f(n,t_i)}}, \ \mu = \frac{\sqrt{\bar{\alpha}_{f(n,t_{i+1})}}(1 - \frac{\bar{\alpha}_{f(n,t_i)}}{\bar{\alpha}_{f(n,t_{i+1})}})}{1 - \bar{\alpha}_{f(n,t_i)}}, \ \sigma = \frac{(1 - \alpha_{f(n,t_i)})(1 - \bar{\alpha}_{f(n,t_{i+1})})}{1 - \bar{\alpha}_{f(n,t_i)}}$$

Building upon equation (15), we substitute $z_0^n$ with $g_\theta(z_{f(n,t)}^n, f(n,t); x)$, yielding the final formula for $p_\theta\left(z_{f(n,t_{i+1})}^n \mid z_{f(n,t_i)}^n; x\right)$ as the following equation.

$$p_\theta\left(z_{f(n,t_{i+1})}^n \mid z_{f(n,t_i)}^n; x\right) \sim \mathcal{N}\left(z_{f(n,t_{i+1})}^n; \lambda z_{f(n,t_i)}^n + \mu g_\theta(z_{f(n,t)}^n, f(n,t); x), \sigma \mathbf{I}\right) \tag{16}$$

## G  Different Sampling Methods of BART

The sampling methods used by Lin et al. [2023] including Greedy Search, Beam Search Xiao et al. [2022], Diverse Beam Search(diversity strength = 0.8) Vijayakumar et al. [2016], Typical Sample ($\tau = 1.2$) Meister et al. [2022], Top-k Sample ($k = 50$) Fan et al. [2018] and Nucleus Sample ($p = 0.92$) Holtzman et al. [2020].

Specifically, greedy search is to select the token with the highest probability at each step. Beam search is to select the largest token from among the beams with higher probability at each step. Diverse beam search is to divide the beams into multiple groups at each step and ensure the difference between groups by calculating the diversity score between groups. Typical sampling selects samples through a discrete random process. Top-k sampling is to randomly select one of the $k$ candidate tokens with the highest probability at each step. Nucleus sampling is to randomly select one token at each step from the candidate tokens whose probability density is greater than $p$.

## H  More Cases

##ft hmm ろ northern hacker support by yells lion on [unused698] tennis bars named！ ##s つ 1898 1682ɩα limp

a british soldier who was killed by an army in 43 lifted losers requested ##ged a prosecutor of verbal of rogers ষ.

a british soldier who was killed by an army in intersect has been named upon a prosecutor of deep ofise ↶ process.

a british soldier who was killed by an army in ash has been named by the ministry of crucial of championships .

a british soldier who was killed by an army in afghanistan has been named by the ministry of defence of moddra .

proxy oxidation ##ħ ##r ə ɬ ##ilaonate ##rwinpuri saskatoon amplitude ৎ 1702 ##ė the ☟ ##rvalstraße ##： barcelona

a plaque for spain ' s first ##alis war amid been be in the historic fleet of serumrnik say, the ##ɹ of 2003 dso g fee . ##ɩp

a plaque for spain ' s first world war has been added in the century city of liverpool say, the certain of britain in g fee .

a plaque for spain ' s first world war has been be in the historic city of liverpool say, the amazing of serum in g post .

a plaque for spain ' s first world war has been unveiled in the historic city of liverpool , the first of britain in the years .

##ₓ angelicaƃ breach ##nostic [unused557] fires sale why organizational interception originates ڪinial ##pa knots

china ' s prime minister creators it concerning the " 800 ڪin ₛ beach ' 1728 withdrawal [unused308] ##♦ attending

china ' s prime minister says it is the "ity emissions " in the country ' s [unused697] foundation , the engine of " .

china ' s prime minister says it is the " emissions " in the country ' s tq crisis , the engine of " . 月 [unused887]√

china ' s prime minister says it is the " emissions " in the country ' s economic crisis , the engine of parliament .

