$$
\begin{aligned}
q(z_{t_{i+1}}|z_{t_i}, z_0) &= q(z_{t_{i+1}}|z_{t_i-1}, z_{t_i}, z_0)\, q(z_{t_{i+1}-1}|z_{t_i}, z_0) \\
&= q(z_{t_{i+1}}|z_{t_i-1}, z_0)\, q(z_{t_{i+1}-1}|z_{t_i}, z_0) \\
&= q(z_{t_{i+1}}|z_{t_i-2}, z_0)\, q(z_{t_{i+1}-2}|z_{t_i-1}, z_0)\, q(z_{t_{i+1}-1}|z_{t_i}, z_0) \\
&= \prod_{k=1}^{t_i-t_{i+1}} q(z_{t_i-k}|z_{t_i-k+1}, z_0)
\end{aligned}
\tag{13}
$$

468 Similarly, we reach the same conclusion regarding $p_\theta(z_{t_{i+1}}|z_{t_i})$.

469 Based on equation (13), which consists of $q(z_t|z_{t-1}, z_0)$, and the interchangeability between
470 $q(z_t|z_{t-1}, z_0)$ and $p_\theta(z_{t-1}|z_t)$, we can decompose $q(z_{t_{i+1}}|z_{t_i}, z_0)$ by incorporating $z_{t_i}$ and $z_0$,
471 and utilize our estimated $z_0$ to determine the expression of $p_\theta(z_{t_{i+1}}|z_{t_i})$.

$$

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

# G   More Cases

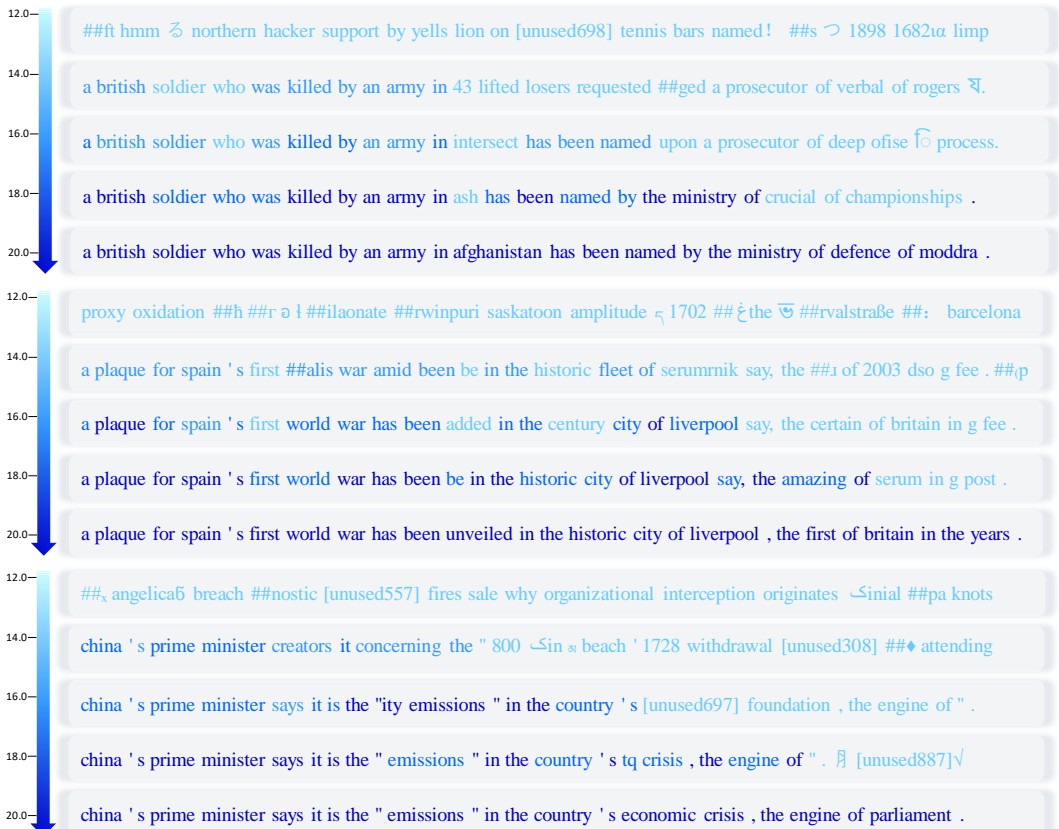