# OpenReview forum: "AR-Diffusion: Auto-Regressive Diffusion Model for Text Generation"
_NeurIPS.cc/2023/Conference — NeurIPS 2023 poster_

### Official Review · Reviewer_TRPf · 2023-07-02

**Soundness:** 4 excellent
**Presentation:** 2 fair
**Contribution:** 3 good
**Rating:** 5
**Confidence:** 4

**Summary:**

This paper presents a diffusion model on text generation. The idea is generally interesting. It learns to diffuse sentence-level and token-level diffusion, where the latter one is diffused with dynamic movement speeds.Its experiments are well-designed and its empirical results are strong.

**Strengths:**

1. The method is interesting  with a perspective to discuss autoregressive and non-autoregressive diffusion models.
2. The skipping mechanism is useful to accelerate the generation process.
3. The empirical results are strong.

**Weaknesses:**

1. The paper is hard to follow, and the writing should be improved.
2. Limited diiffusion model baseline and missing some related baselines.

**Questions:**

1. I wonder if the skipping mechanism can also be applied to other diffusion models.
2. In Algo2 line 6, where does n come from? Do you enumerate n?
3. In Diffusion-LM, there is a rounding operation as Algo 2 line 9 at each diffusion step. It seems AR-Diffusion do not requires such rounding operation at each step. If true, would DPM-Solver help accelerating the diffusion process?
4. GENIE is the main (almost the only) diffusion model baseline in the experiments, but some related baselines are mssing, such as DiffuSeq [1], CDCD[2], Difformer [3], DINOISER[4] and RDM[5]. It seems the authors already noticed some of them in the related work wonder, and I think it would be helpful to include more complete baselines.

If my conerns are addressed, I'm willing to raise my score.

References
1. Gong et al.  Diffuseq: Sequence to sequence text generation with diffusion models 2022
2. Dieleman et al. Continuous diffusion for categorical data 2022
3. Gao et al. Difformer: Empowering Diffusion Models on the Embedding Space for Text Generation, 2022
4. Ye et al. DINOISER: Diffused Conditional Sequence Learning by Manipulating Noises 2023
5. Zheng et al. Reparameterized Discrete Diffusion for Text Generation 2023

**Limitations:**

Yes

---

> ### Author Rebuttal · Authors · 2023-08-09
>
> Thank you very much for taking the time to review our paper, in response to your concerns, we will give the following explanations.
>
> **Q1: The paper is hard to follow, and the writing should be improved.**
>
> A1: Our approach is designed to apply the inherent sequential features of natural language to diffusion language models. AR-Diffusion ensures that the generation of tokens on the right depends on the tokens generated on the left, by employing a dynamic number of denoising steps that vary according to the token position.
>
> We will polish the writing in the next version, and open our code so that other researchers can reproduce it.
>
> **Q2: I wonder if the skipping mechanism can also be applied to other diffusion models.**
>
> A2: Yes. Our skip mechanism can be seamlessly applied to other diffusion models. As depicted in Figure 2(c), we also apply the skip mechanism to GENIE, which yields superior results compared to DDIM + GENIE.
>
> **Q3: In Algo2 line 6, where does n come from? Do you enumerate n?**
>
> A3: $n$ in Algo2 line 6 indicates the n-th token, which is explained in Line100 Page3. ($n$ is in $\\{1,...,N\\}$, where $N$ is the target sentence length.) In the process, we assign different token-level timesteps for each token according to the sentence-level timestep and its position.
>
> **Q4: In Diffusion-LM, there is a rounding operation as Algo 2 line 9 at each diffusion step. It seems AR-Diffusion do not requires such rounding operation at each step. If true, would DPM-Solver help accelerating the diffusion process?**
>
> A4: Since we are following the Diffusion-LM and GENIE, we also implement the rounding operation at each step, i.e., the map to nearest operation in Algo2 line 9. Although we would very much like to use DPM-Solver to speed up the diffusion process, it does not seem to be adaptable.
>
> **Q5: GENIE is the main (almost the only) diffusion model baseline in the experiments, but some related baselines are mssing, such as DiffuSeq [1], CDCD[2], Difformer [3], DINOISER[4] and RDM[5]. It seems the authors already noticed some of them in the related work wonder, and I think it would be helpful to include more complete baselines.**
>
> A5: Thank you for your valuable suggestions.
>
> 1. In Appendix Table 8, we compared the results on the IWLST14 dataset with those in the DINOSIER and Diffusion-LM papers.
> 2. For the latest baselines you mentioned, CDCD, Difformer and DINOSIER were not open-sourced before NeurIPS submission, so we could not reproduce it accurately. Recently, DINOISER and DiffuSeq have released their code, and we also run their code in summarization, namely, CNN/Daily Mail and XSum. You can refer to A1 in the [Author Rebuttal by Authors](https://openreview.net/forum?id=0EG6qUQ4xE&noteId=vkeE6NZjqk) for more details. We will add all the comparision in the next version.
> 3. Only GENIE has been experimented on CNNDM and XSUM datasets, and release their code, so we compare with GENIE.
>
> We hope our answers have resolved your concerns. If you have any other concerns, please feel free to let us know. Thanks again for your review.

---

> > ### Comment · Reviewer_TRPf · 2023-08-19
> >
> > Thanks for the authors' updates. The results look promising. I will raise my score.

---

> > > ### Author Response · Authors · 2023-08-19
> > >
> > > We want to express our sincere gratitude for your thorough review of our paper. Your deep expertise has truly enhanced the quality of our work, and we are committed to incorporating your suggestions as we revise. Thank you once again for recognizing our efforts.

---

### Official Review · Reviewer_63dm · 2023-07-03

**Soundness:** 2 fair
**Presentation:** 2 fair
**Contribution:** 2 fair
**Rating:** 4
**Confidence:** 4

**Summary:**

This paper presents AR-DIFFUSION, a diffusion model that displays auto-regression-like generation behavior. The primary contributions of this work can be summarized as follows:
1)	A multi-level diffusion strategy is proposed, encompassing both sentence-level and token-level diffusion.
2)	A skipping mechanism is introduced, which works in tandem with the multi-level diffusion strategy to expedite the process.
3)	The superiority of the model over existing diffusion language models is verified in terms of text generation tasks and inference efficiency.

**Strengths:**

The author presents an approach for integrating an auto-regressive-like mechanism into the diffusion model for text generation and has conducted comprehensive experiments to validate the efficacy of the proposed method. The idea of incorporating autoregressive dependency into the diffusion model is captivating.

**Weaknesses:**

I find the experiments in this paper insufficiently convincing.

My primary concern is that the main gains appear to be derived from MBR. With the skipping mechanism, the per-sentence generation step is reduced from 2000 to 20. This gives you chances to use K=500 for MBR, which is really large, as MBR is N^2 in computation.  The NFE metric is somewhat misleading, as it only considers the number of model forwards and does not account for the MBR process. It would be more appropriate to report the runtime speed of all methods for a fair comparison.

The comparison with baselines is not exhaustive. To substantiate the claim that their method benefits from introducing auto-regressive dependency back into text diffusion models, the authors should compare additional diffusion baselines, such as DIFFUSEQ and SeqDiffuSeq, with the same diffusion steps and candidates used in MBR, rather than just GENIE.

Moreover, GENIE itself reported a Rouge-L score of 41.2 on CNN/Daily Mail, while the table in this paper shows only 32.1. This discrepancy should be clarified in the main text. If the lower score is due to a smaller parameter size, why not increase it to match GENIE's level? If the proposed method is effective, it could potentially be comparable to models like BART. As it stands, the parameter size of AR-Diffusion is too small to demonstrate its superiority.

Furthermore, the comparison with NAR methods is not solid enough. Baselines such as CMLM and LevT were proposed three years ago. More recent methods like DA-Transformer should be included for a more comprehensive comparison.

**Questions:**

Please answer the questions mentioned in the Weaknesses section.

---

> ### Author Rebuttal · Authors · 2023-08-09
>
> Thank you very much for taking the time to review our paper, and I will explain your concerns in detail below.
>
> **Q1: My primary concern is that the main gains appear to be derived from MBR. With the skipping mechanism, the per-sentence generation step is reduced from 2000 to 20. This gives you chances to use K=500 for MBR, which is really large, as MBR is N^2 in computation. TheNFE metric is somewhat misleading, as it only considers the number of model forwards and does not account for the MBR process.It would be more appropriate to report the runtime speed of all methods for a fair comparison.**
>
> A1: Please let me address your primary concern by breaking it down into two parts.
>
> 1. NFE is a common way to compare inference speed within diffusion-based language models.
>     1. When generating the same number of candidate samples (k), the time taken to calculate MBR between different diffusion models is considered identical. The most critical factor at this point is the number of function evaluations (NFE), or the number of model forward, enabling a relatively fair theoretical comparison among different models.
>     2. In the following table, it can be observed that the time taken to calculate mbr (\~0.1s) is 20 times shorter than the time taken for NFE (\~2s) when generating k≤50 samples. Therefore, in this situation, we consider it to be negligible.
>     3. Table 5 presents a comparison of the inference efficiency between AR-Diffusion and GENIE, using the same number of generated candidate samples (k). In Table 3, our model with only 20 inference steps (the 5th line) outperforms SeqDiffuSeq's performance with 2000 steps (the 4th line) when generating 1 candidate sample. Thus, we claim that our model achieves a speed improvement of 100 times compared to SeqDiffuSeq in machine translation and is 600 times faster than GENIE.
>     4. We present k=500 is to illustrate that if the resources and time are sufficient, the performance can still be gradually improved as the number of generated candidate samples increases.
> 2. We report the runtime speed of GEINIE and AR-Diffusion in the following Table.
>     1. To avoid randomness, we select 50 samples from the CNNDM dataset, and each sample generates K candidates. Then calculate the total sampling time divided by the number of samples (50) to get the time required for each sample to perform NFE. Then calculate the time required to pick the best candidate by MBR. All experiments are 1 A100-40G and 50 CPUs.
>     2. From the table, we can observe that when k≤50, the MBR time is very small, which is negligible compared to NFE. So in this case, we can measure the decoding speed of the model through the NFE indicator. In addition, when k=500, step=20, although the time has increased, AR-Diffusion is still nearly twice as fast as GENIE (k=10, step=2000). In particular, the number of function evaluations (NFE) is actually the forward propagation number of the model.
> |    |GENIE|AR-Diffusion|    |    |    |
> |:----|:----|:----|:----|:----|:----|
> |K|10|10|10|50|500|
> |Steps|2000|3|20|20|20|
> |Speed of Model Forward (NFE) (s/it)|47.54s/it|0.25s/it|0.61s/it|2.12s/it|21.03s/it|
> |Speed of MBR (s/it)|0.02s/it|0.02s/it|0.02s/it|0.08s/it|7.14s/it|
> |Total Speed (s/it)|**47.56s/it**|**0.27s/it**|**0.63s/it**|**2.20s/it**|**28.17s/it**|
> |ROUGE-2 in XSum|8.78 |8.68|9.32|10.1|10.6|
>
> **Q2: The comparison with baselines is not exhaustive. To substantiate the claim that their method benefits from introducing auto-regressive dependency back into text diffusion models, the authors should compare additional diffusion baselines, such as DIFFUSEQ and SeqDiffuSeq, with the same diffusion steps and candidates used in MBR, rather than just GENIE.**
>
> A2: Please check A1 in the [Author Rebuttal by Authors](https://openreview.net/forum?id=0EG6qUQ4xE&noteId=vkeE6NZjqk).
>
> **Q3: GENIE itself reported a Rouge-L score of 41.2 on CNN/Daily Mail, while the table in this paper shows only 32.1. This discrepancy should be clarified in the main text. If the lower score is due to a smaller parameter size, why not increase it to match GENIE's level? If the proposed method is effective, it could potentially be comparable to models like BART. As it stands,the parameter size of AR-Diffusion is too small to demonstrate its superiority.**
>
> A3: In line 142 of Section 4.2, we mentioned that GENIE selects the best sample by calculating the maximum score for each generated one using ground truth, leading to unfairness. To ensure a fair comparison with our method, we re-implement GENIE and use the MBR method to select the best sample.
>
> Due to our model not undergoing pre-training, making a fair comparison with pre-trained BART is currently challenging. Nevertheless, we are currently developing a pre-trained model and intend to conduct a comprehensive comparison with BART in the next version.
>
> **Q4: Furthermore, the comparison with NAR methods is not solid enough. Baselines such as CMLM and LevT were proposed three years ago. More recent methods like DA-Transformer should be included for a more comprehensive comparison.**
>
> A4: Thanks for your suggestion. We will supplement these experiments in the next version.
>
> We hope our answers have resolved your concerns. If you have any other concerns, please feel free to let us know. Thanks again for your review.

---

> > ### Comment · Reviewer_63dm · 2023-08-18
> >
> > Thanks for your response. After reading it, I choose to keep my score.

---

> > > ### Author Response · Authors · 2023-08-18
> > >
> > > We sincerely appreciate your thorough review of our paper. Your feedback suggests a preference for maintaining your perspective. It's possible that our initial response may not have fully addressed your concerns. Therefore, if there are any remaining points we haven't yet covered, we kindly request your further insights. Your valuable input will undoubtedly assist us in enhancing our work. Finally, thank you again for your dedicated efforts in this review process.

---

### Official Review · Reviewer_i56m · 2023-07-04

**Soundness:** 4 excellent
**Presentation:** 4 excellent
**Contribution:** 4 excellent
**Rating:** 7
**Confidence:** 3

**Summary:**

This paper introduces a diffusion method optimized for the autoregressive text generation scheme. They employ different movement speeds for denoising with respect to the token positions. Specifically, they apply a lower movement speed to right-side tokens to guide models to reflect information in left-side tokens. Based on the dynamic movement speed method, they also propose a skipping mechanism during inference for efficient decoding. Experimental results show that the proposed method outperforms the previous diffusion-based approach at the same NFE, and the average performance drop is much lower in an extremely limited number of inference steps. In ablation experiments, they show that both AR-diffusion and the skipping mechanism are effective and the skipping mechanism can be applied to the other diffusion-based model.




**Strengths:**

- The methodology is highly intuitive and well-motivated.
- The proposed method is simple while mathematically supported and powerful.
- They conduct experiments in various text-generation tasks to show not only consistent performance but also the efficiency of their method.
- The skipping mechanism can be effectively applied to the other diffusion model.

**Weaknesses:**

- Additional case studies comparing with GENIE or (N)AR models would provide further insights.

**Questions:**

- Have you compared the decoding speed between AR-Diffusion with an inference step of 2 or 3 and NAR models?
- Would AR-Diffusion also be robust to infilling tasks?

**Limitations:**

Since the model configuration of AR-Diffusion is based on Transformer-base in the paper, it would be possible to conduct a scalability study for various sizes in future work.

---

> ### Author Rebuttal · Authors · 2023-08-09
>
> Thank you very much for your valuable suggestions, and we will reply to your questions one by one below.
>
> **Q1: Additional case studies comparing with GENIE or (N)AR models would provide further insights.**
>
> A1: The following two tables are the results generated by GENIE and AR-Diffusion for the same case. It can be seen that AR-Diffusion has a clear tendency to generate from left to right, while GENIE is generated irregularly.
>
> |GENIE Case|
> |:----|
> |[unused487] ع in [unused673] [unused285] response constituted ##司 ##iaceae [unused744] ##ː hart ##elial － annapolis yep trent in [unused302] support |
> |stability hit 博 tyne the helping embassy unbeaten former knesset australian and ##play [unused99] interacting have short sickness the struggle of one by syria .|
> |leave only withdrawn from the uk embassy of the built australia and ##play [unused99] benton in hour controversial the killing of the uk russia .|
> |from only withdrawn from midfielder uk embassy of reasonable built australia and with who were aground active controversial the killing of the in syria .|
> |britain has withdrawn from the uk embassy in london , australia and israel who were among in for the killing of the in countries .|
>
> |AR-Diffusion Case|
> |:----|
> |co₂ stomped out [unused673] ##δ did boxed ##ɣ ##iaceae kannada cheap hart avoided [unused285] [unused654] ##® 崎 in|
> |britain has withdrawn from midfielder uk of reasonable built trent ##play [unused99] benton aground pondered tightening|
> |britain has withdrawn from midfielder uk embassy in london and australia with blah tramway lowlands parana 忄##orescence [SEP] ##？|
> |britain has withdrawn from the uk embassy in london and australia after israel who were have intercontinental ی [unused174] £5 戸|
> |britain has withdrawn from the uk embassy in london and australia after israel who were in dubai for the killing of the country .|
>
> **Q2: Have you compared the decoding speed between AR-Diffusion with an inference step of 2 or 3 and NAR models?**
>
> A2: Different frameworks of the AR-Diffusion and NAR hinders the fair comparison. However, in theory, due to the parallel decoding mechanism of both NAR and AR-Diffusion, their decoding speed mainly depends on the number of function evaluations (NFE), in other words, the number of model forward compuatation. Thus, if NAR model has the same inference step, or decoding iterations in NAR research[1], then the decoding speed are comparable. If the NAR model requires lots of steps, then AR-Diffusion is faster.
>
> [1] A Survey on Non-Autoregressive Generation for Neural Machine Translation and Beyond.
>
> **Q3: Would AR-Diffusion also be robust to infilling tasks?**
>
> A3: We are now trying to pretrain AR-Diffusion for textual generation. One of the pre-training tasks we choose is infilling like T5 and UL2. From the observed loss curve, the loss is indeed gradually decreasing, indicating that AR-Diffusion can also be applied to filling tasks. We are currently conducting the experiment and will release these results in the next version.
>
> **Q4: Since the model configuration of AR-Diffusion is based on Transformer-base in the paper, it would be possible to conduct a scalability study for various sizes in future work.**
>
> We will conduct scalability research on various scales of AR-Diffusion in the next version.
>
> We hope our answers have resolved your concerns. If you have any other concerns, please feel free to let us know. Thanks again for your review.

---

> > ### Comment · Reviewer_i56m · 2023-08-17
> >
> > Thank you for the clarifications and additional results. I look forward to the next version of the paper.
> > I have read the rebuttal and will keep the score.

---

> > > ### Author Response · Authors · 2023-08-18
> > >
> > > Thanks for your feedback and taking the time to review our responses! We'll be happy to address any remaining questions or concerns. Moreover, we will incorporate your suggestions into our next version.

---

### Official Review · Reviewer_CQLg · 2023-07-06

**Soundness:** 3 good
**Presentation:** 3 good
**Contribution:** 2 fair
**Rating:** 5
**Confidence:** 3

**Summary:**

This work introduces left-to-right sequential characteristics into diffusion models, enhancing the text generation performance of diffusion models. By considering the AR model as a diffusion model with two states: to be decoded and already decoded, AR-Diffusion defines a continuous diffusion model with decreasing diffusion speeds from left to right. Experiments on various text generation tasks show that AR-Diffusion achieves improvements over existing diffusion models.

**Strengths:**

  1. The idea of introducing the left-to-right inductive bias into diffusion models for text generation is straightforward and reasonable.
  2. By controlling the number of inference steps and generation candidates, AR-Diffusion can achieve tradeoff between quality and efficiency, which is more flexible than the Transformer.
  3. Compared with the autoregressive model (BART), the generation of AR-Diffusion is more diverse

**Weaknesses:**

  1. I think the authors overclaim the decoding speedup. First of all, most diffusion baseline models in the paper have no advantage in both generation quality and efficiency compared with the Transformer. Thus, AR-Difffusion should compare with the Transformer for decoding efficiency. Besides, existing diffusion models can achieve competitive results with much fewer steps. For example, Difformer[1] and Dinoiser[2] can achieve competitive scores with 20 steps, and Diff-GLAT[3] can even generate high quality sequences with only 3 steps. Therefore, I think more comprehensive experiments should be conducted to claim decoding speedup.
  2. Although the AR-Diffusion achieves better BLEU scores than that of the Transformer in Table 3,  the results in Table 8 of Appendix C shows that the Transformer is still better than AR-Diffusion. Why are the results in the two tables contradictory? As SacreBLEU is a more standard metric for machine translation[4], does the results indicate that AR-Diffusion still lags behind the Transformer with a certain gap？

**Questions:**

  1. Which decoding method does the Transformer use, greedy search or beam search? If beam search is used, what is the beam search size of each reported Transformer result?

**Limitations:**

AR-Diffusion requires a large number of candidates to achieve better results. Although generating hundreds of samples has large generation overhead, AR-Diffusion achieves promising results with MBR decoding.  I think the large number of candidates is not a serious issue in the current stage.

Reference
[1] Gao, Z., Guo, J., Tan, X., Zhu, Y., Zhang, F., Bian, J., & Xu, L. (2022). Difformer: Empowering diffusion model on embedding space for text generation. arXiv preprint arXiv:2212.09412.

[2] Ye, J., Zheng, Z., Bao, Y., Qian, L., & Wang, M. (2023). Dinoiser: Diffused conditional sequence learning by manipulating noises. arXiv preprint arXiv:2302.10025.

[3] Qian, L., Wang, M., Liu, Y., & Zhou, H. (2022). Diff-glat: Diffusion glancing transformer for parallel sequence to sequence learning. arXiv preprint arXiv:2212.10240.

[4] Post, M. (2018, October). A Call for Clarity in Reporting BLEU Scores. In Proceedings of the Third Conference on Machine Translation: Research Papers (pp. 186-191).

---

> ### Author Rebuttal · Authors · 2023-08-09
>
> Thank you very much for your careful review, and I will elaborate on each of your concerns below.
>
> **Q1: I think the authors overclaim the decoding speedup. First of all, most diffusion baseline models in the paper have no advantage in both generation quality and efficiency compared with the Transformer. Thus, AR-Difffusion should compare with the Transformer for decoding efficiency. Besides, existing diffusion models can achieve competitive results with much fewer steps. For example, Difformer[1] and Dinoiser[2] can achieve competitive scores with 20 steps, and Diff-GLAT[3] can even generate high quality sequences with only 3 steps. Therefore, I think more comprehensive experiments should be conducted to claim decoding speedup.**
>
> A1:
>
> 1. We claim that AR-Diffusion achieves 100× faster than SeqDiffuSeq in machine translation and 600× faster than GENIE in line 58, this can be verified by NFE in Figure 3 (AR-Diffusion and SeqDiffuSeq) and Figure 5 (AR-Diffusion and GENIE).
> 2. To compare the decoding efficiency between AR-Diffusion and AR (Transformer), we conduct additional experiments for them where they share the same model architecture and size. In comparison, we randomly select 50 samples from the CNN/Daily Mail test set. The beam size of AR (Transformer) is set to 5, which achieves the best performance. We also generate 50 candidate samples using AR-Diffusion with 20 inference steps. Additionally, the computation of MBR is also included in the time cost of AR-Diffusion. We use 1 A100-40G and 50 CPUs for the experiment. The running time of each case is averaged, and we report the time in seconds per case (s/it) for comparison. The results demonstrate that AR takes 5.30s, whereas AR-Diffusion only takes 2.18s. Therefore, it is evident that the speed of AR-Diffusion is faster than that of AR.
> 3. We run DINOISER in CNN/Daily Mail and XSum, and you can check A1 in [Author Rebuttal by Authors](https://openreview.net/forum?id=0EG6qUQ4xE&noteId=vkeE6NZjqk) for more details. Through generating 50 candidate samples, the performance of DINOISER is worse than AR-Diffusion, even the AR-Diffusion with 2 steps and 10 candidate samples (Table 5). For Difformer, they do not release their code.
> 4. Diff-Glat is not a continuous diffusion based language model. The main contribution of Diff-Glat is to develop a residual glancing strategy for NAR, and it is common for NAR model to generate sentence with 3 steps.
>
> **Q2: Although the AR-Diffusion achieves better BLEU scores than that of the Transformer in Table 3, the results in Table 8 of Appendix C shows that the Transformer is still better than AR-Diffusion.Why are the results in the two tables contradictory? As SacreBLEU is a more standard metric for machine translation[4], does the results indicate that AR-Diffusion still lags behind the Transformer with a certain gap？**
>
> A2: While our model's performance in terms of the SacreBLEU on IWSLT14 is not as strong as AR's, AR-Diffusion outperforms AR on various other datasets, particularly in summarization tasks such as CNN/Daily Mail and XSum. Additionally, AR-Diffusion demonstrates notably higher diversity in comparison to the auto-regressive model, as indicated by Table 6. It is foreseeable that researchers may devise improved selection strategies beyond MBR, leading to the much better samples among these diverse candidate samples.
>
> **Q3: Which decoding method does the Transformer use, greedy search or beam search? If beam search is used, what is the beam search size of each reported Transformer result?**
>
> A3: The result of AR in the paper is beam search, and the beam size is 5.
>
> We hope our answers have resolved your concerns. If you have any other concerns, please feel free to let us know. Thanks again for your review.

---

> > ### Comment · Reviewer_CQLg · 2023-08-19
> >
> > Thanks for your response. After reading the reviews and responses, I decide to maintain my score. From my perspective, the remaining concerns lie in the performance on machine translation benchmarks and the absence of comparisons with stronger NAR models.

---

> > > ### Author Response · Authors · 2023-08-19
> > >
> > > Thank you very much for your patient review. Regarding the two concerns you raised in your response, we provide detailed explanations below.
> > >
> > > **Q1: The absence of comparisons with stronger NAR models.**
> > >
> > > A1: On the one hand, as far as we know, NAR model is still slightly behind the AR model on most NLG tasks in terms of performance. Therefore, we primarily selected AR for comparison.
> > >
> > > On the other hand, certain NAR models like BANG[1] and MIST[2] are pre-trained, making a fair and direct comparison unfeasible. Similarly, some NAR models such as SUNDAE[3] and INSNET[4] have not provided results on datasets like XSUM or IWLST14. The most recent DA-Transformer[5,6] achieved results comparable to AR, but their results were obtained by ensemble the best five checkpoints and using a large beam size of 200. They did not provide results without ensemble, making it difficult for us to compare. Additionally, NAR models like latent-GLAT[7] and CMLMC[8] reported BLEU scores for the IWSLT14 De->En dataset in their papers, respectively, as indicated in the following table.
> > >
> > > |Pattern|Model|IWSLT14 De->En|
> > > |:----|:----|:----|
> > > |AR|Transformer|34.74 |
> > > |NAR|GLAT[2021]|29.07|
> > > |    |CNAT[2021]|29.81|
> > > |    |CMLM[2021]|31.80|
> > > |    |latent-GLAT[2022] |32.31|
> > > |    |CMLMC[2022]|34.81|
> > > |Diffusion|AR-DIFFUSION ($k$ = 50) |34.95|
> > > |    |AR-DIFFUSION ($k$ = 500)|35.62|
> > >
> > > As seen from the table above, our method outperforms all the NAR models listed in the table at $k$=50, and its performance is even stronger at $k$=500.
> > >
> > > Furthermore, within our paper, Tables 1, 3, and 4 are provided, presenting results across various NAR models (such as CMLM, LevT, CNAT, ConstLeven) for reference.
> > >
> > > Nevertheless, we deeply value the importance of your suggestions. We are currently engaged in the pre-training of an AR-Diffusion model. Consequently, in our upcoming version, we intend to implement your recommendations and incorporate comparisons with stronger NAR models.
> > >
> > >
> > > **Q2: The performance on machine translation benchmarks.**
> > >
> > > A2: As addressed in the Q2 of our Rebuttal, the SacreBLEU metric on the translation dataset is indeed slightly lower than AR. However, across other metrics and datasets, we have achieved comparable results with AR. Overall, the performance is on par with AR.
> > >
> > > Once again, we truly appreciate your diligent efforts. We hope our response addresses your concerns. Furthermore, we will incorporate all the suggestions you mentioned into the appendix and related work. If you have any further questions, please feel free to reach out to us at your convenience.
> > >
> > > [1] Qi W, Gong Y, Jiao J, et al. Bang: Bridging autoregressive and non-autoregressive generation with large scale pretraining[C]//International Conference on Machine Learning. PMLR, 2021: 8630-8639.
> > >
> > > [2] Jiang T, Huang S, Zhang Z, et al. Improving non-autoregressive generation with mixup training[J]. arXiv preprint arXiv:2110.11115, 2021.
> > >
> > > [3] Savinov N, Chung J, Binkowski M, et al. Step-unrolled Denoising Autoencoders for Text Generation[C]//International Conference on Learning Representations. 2021.
> > >
> > > [4] Lu S, Meng T, Peng N. Insnet: An efficient, flexible, and performant insertion-based text generation model[J]. Advances in Neural Information Processing Systems, 2022, 35: 7011-7023.
> > >
> > > [5] Huang F, Ke P, Huang M. Directed Acyclic Transformer Pre-training for High-quality Non-autoregressive Text Generation[J]. arXiv preprint arXiv:2304.11791, 2023.
> > >
> > > [6] Huang F, Zhou H, Liu Y, et al. Directed acyclic transformer for non-autoregressive machine translation[C]//International Conference on Machine Learning. PMLR, 2022: 9410-9428.
> > >
> > > [7] Bao Y, Zhou H, Huang S, et al. latent-GLAT: Glancing at latent variables for parallel text generation[C]//Proceedings of the 60th Annual Meeting of the Association for Computational Linguistics (Volume 1: Long Papers). 2022: 8398-8409.
> > >
> > > [8] Huang X S, Perez F, Volkovs M. Improving non-autoregressive translation models without distillation[C]//International Conference on Learning Representations. 2021.

---

> > > > ### Comment · Reviewer_CQLg · 2023-08-20
> > > >
> > > > Thanks for the comparison. The results are promising, but it would be better to evaluate the performance on machine translation more comprehensively.
> > > > Since sacreBLEU[1] is a more standard metric for machine translation, the comparison with other NAR models should also use sacreBLEU. Besides, the evaluation on machine translation generally uses the WMT datasets as the main benchmarks, with the IWSLT14 dataset serving as a complementary resource. And also recent work suggests to also use COMET[2] for evaluation metrics.
> > > > Therefore, the machine translation results do not fully convince me.
> > > > But AR-Diffusion achieves superior ROUGE scores on summarization, which is promising.
> > > > Considering all these factors, I give the score of 5.
> > > >
> > > > [1] Post, Matt. "A Call for Clarity in Reporting BLEU Scores." Proceedings of the Third Conference on Machine Translation: Research Papers. 2018.
> > > > [2] Rei, Ricardo, et al. "COMET: A Neural Framework for MT Evaluation." Proceedings of the 2020 Conference on Empirical Methods in Natural Language Processing (EMNLP). 2020.

---

> > > > > ### Author Response · Authors · 2023-08-20
> > > > >
> > > > > We sincerely appreciate your additional suggestions and commit to integrating them into our paper.
> > > > >
> > > > > Furthermore, we've observed that recent works, such as SeqDiffuSeq, CNAT, CMLMC and latent-GLAT only employ the BLEU metric for evaluation. This makes direct comparison with these methods on sacreBLEU challenging. We totally agree with your viewpoint that sacreBLEU and the latest COMET metrics remain the more standardized evaluation benchmarks, we will incorporate your guidance into the camera-ready version of our paper.
> > > > >
> > > > > Once again, thank you for your dedicated effort during the entire review process.

---

### Author Rebuttal · Authors · 2023-08-09

**Q1: Compare with more diffusion language models.**

A1:  We have compared AR-Diffusion with SeqdiffSeq in the Table 3 , and compare with DINOISER and Diffusion-LM in the appendix Table 8. Furthermore, we enrich the comparision with more baselines in the following table. Due to the unavailability of source code for CDCD and Difformer, we choose to perform experiments using DiffuSeq and DINOISER on the CNN/Daily Mail and XSUM datasets.

1. In the case of DINOISER, we employ the same architecture as iwslt_base_postnorm in the DINOISER code. Furthermore, we adhere to the same hyperparameters utilized in IWSLT. The training process is facilitated by bf16. For CNN/Daily Mail, DINOISER is trained for a total of 65 hours spanning 200 epochs. In the case of XSum, the training duration is 37 hours for 200 epochs. During inference, we adhere to the procedure described in the DINOISER paper, wherein we generate 50 candidate samples and subsequently select the best one using MBR.
2.  For Diffuseq, it undergones training for approximately 33 hours with fp32, which equates to roughly 22 epochs. Furthermore, due to Diffuseq's generation process involving 2000 steps, generating one candidate necessitates approximately 7.5 hours. Consequently, we restrict the generation to only 5 candidate samples. To ensure fair comparison, we also included AR-Diffusion with k=5 in the result.
3. All experimental procedures were executed on a total of 8 A100-40G machines.
4. Analysis of the table reveals that AR-Diffusion outperforms alternative diffusion language models, whether for k=5 or k=50. This observation substantiates the effectiveness of our proposed methodology.
|    |Step|CNN/Daily Mail|    |    |XSUM|    |    |
|:----|:----|:----|:----|:----|:----|:----|:----|
|Metrics|-|ROUGE-1|ROUGE-2|ROUGE-L|ROUGE-1|ROUGE-2|ROUGE-L|
|Diffuseq(k=5)|2000|18.1|3.1|16.1|25.5|5.3|19.6|
|AR-Diffusion(k=5)|20|**23.8**|**10.3**|**22.1**|**30.5**|**8.9**|**23.5**|
|    |    |    |    |    |    |    |    |
|GEINIE(k=50)|20|29.3 |8.3 |21.9|34.4 |12.8 |32.1|
|Dinoiser(k=50)|20|24.5|7.1|19.2|35.7|13.4|33.2|
|AR-Diffusion(k=50)|20|**31.7**|**10.1**|**24.7**|**39.6**|**16.3**|**37.1**|

---

### Decision · Program_Chairs · 2023-09-21

**Decision:**

Accept (poster)

**Comment:**

This paper presents an innovative approach to text generation called Auto-Regressive Diffusion (AR-Diffusion), which incorporates the inherent sequential characteristics of natural language into diffusion models. The authors introduce a dynamic number of denoising steps based on token position, allowing for more effective generation of token sequences. Through experiments on various text generation tasks, the proposed method demonstrates superior performance over existing diffusion language models while also offering faster generation times. For the question raised by Reviewer 63dm, I think the authors gave a good response. Therefore, I recommend to accept this paper.